# SKETCHKNITTER: VECTORIZED SKETCH GENERATION WITH DIFFUSION MODELS

**Qiang Wang**[1]    **Haoge Deng**[1]    **Yonggang Qi**[1*]    **Da Li**[2]    **Yi-Zhe Song**[2]

[1]Beijing University of Posts and Telecommunications, CN    [2]SketchX, CVSSP, University of Surrey, UK

{*wanqqiang, denghaoge, qiyg*}*@bupt.edu.cn*    *dali.academic@gmail.com*    *y.song@surrey.ac.uk*

## ABSTRACT

We show vectorized sketch generation can be identified as a reversal of the stroke deformation process. This relationship was established by means of a diffusion model that learns data distributions over the stroke-point locations and pen states of real human sketches. Given randomly scattered stroke-points, sketch generation becomes a process of deformation-based denoising, where the generator rectifies positions of stroke points at each timestep to converge at a recognizable sketch. A key innovation was to embed *recognizability* into the reverse time diffusion process. It was observed that the estimated noise during the reversal process is strongly correlated with sketch classification accuracy. An auxiliary recurrent neural network (RNN) was consequently used to quantify recognizability during data sampling. It follows that, based on the recognizability scores, a sampling shortcut function can also be devised that renders better quality sketches with fewer sampling steps. Finally it is shown that the model can be easily extended to a conditional generation framework, where given incomplete and unfaithful sketches, it yields one that is more visually appealing and with higher recognizability.

## 1 INTRODUCTION

Free-hand human sketches are abstract concepts which can efficiently express ideas. Generative models for sketches have received increasing attentions in recent years. Compared with producing pixelated sketches (Ge et al., 2020; Chen et al., 2001; Liu et al., 2020), modeling sketches with point trajectories is more reasonable and appealing as it more closely resembles drawing process of humans. Sketch-RNN (Ha & Eck, 2018) utilizes a set of discrete stroke points and binary pen states as an approximation of the continuous drawing trajectory. BézierSketch (Das et al., 2020) makes use of parametric representation, which fits the stroke trajectory by Bézier curves. Very recently, SketchODE (Das et al., 2021a) applies neural ordinary differential equations to representing stroke trajectory through continuous-time functions. All said approaches however suffer from the inability to model complex vectorized sketches. This is largely attributed to the de-facto RNN backbone that falls short in accommodating large stroke point numbers – rule of thumb is anything beyond 200 points will fail (Pascanu et al., 2013; Das et al., 2021b).

In this paper, we attempt to change the status quo in how stroke-point trajectories are modeled. Instead of seeing sketch generation as a process of determining where the next stroke-point lies under each recurrent step (as per RNN), we attempt to estimate distributions of *all* stroke-points holistically at each time instance – as every knitting enthusiast will tell you, it is all about having a global plan, never just about the next thread! [1].

Our key novelty lies with the realization that sketch generation can be conceptualized as the reversal of a stroke deformation process. Through modeling a forward deformation process (i.e., sketch to noise), our diffusion model learns the stroke-point distributions of real human sketches, and thus able to reverse the process to generate novel sketches given noisy input. It follows that given this diffusion setup, the sequential information in sketches can be persevered by simply maintaining the temporal ordering of stroke-points during reverse-time diffusion.

---

*Correspondence to: Yonggang Qi (qiyg@bupt.edu.cn). Code to be found at GitHub page

[1]`https://www.ravelry.com/`

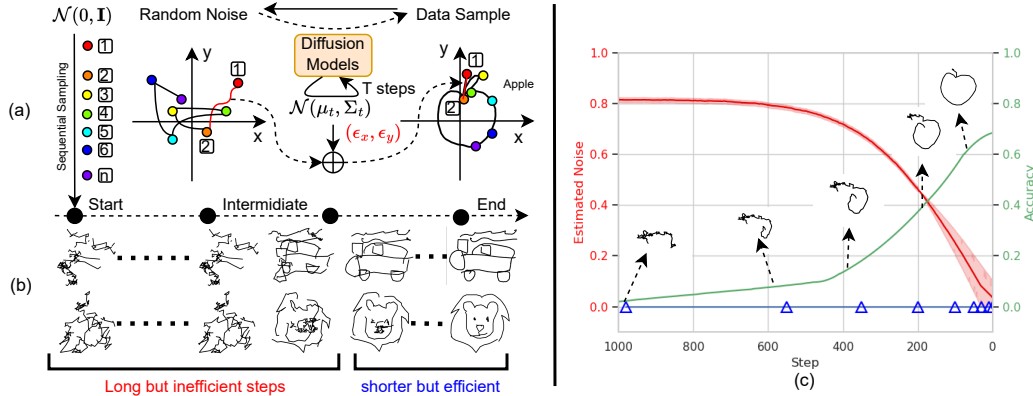

Figure 1: (a) Built on diffusion models, sketch generation is formulated as a stroke deformation rectification problem. Essentially, our model is to reorganize points with fixed adjacency, such that placing them on meaningful locations out of a total mess, dubbed sketch knitting. Note that the order of stroke points is pre-determined and unchanged. (b) Generated sketches with different deformation levels associated with timesteps by our model. Early sampling stage is not efficient that the obtained data change little and remain noisy. (c) Early sampling is inefficient as basically the same estimated noise to each stroke point is used, led sketch remains noisy and unrecognizable. And different noise patterns can be observed at different timesteps, which motivate us to devise recognizability-based skipping (blue△) based on the estimated noise to find shortcut sampling. Red solid curve denotes mean and the shade denotes variance. 1000 generated sketches are used for plotting.

We further draw importance on the overall quality (recognizability) of the sketches generated. We show that the estimated noise in the sampling stage naturally can reflect the recognizability of the generated sketch at each timestep. It follows that a learnable RNN was devised to explicitly model the relation between estimated noise and recognizability. This is achieved by introducing an pre-trained image classifier as supervision signal. Embedding recognizabilty into the sampling process also yields the added benefit of introducing skip steps that allows for more efficient and effective data generation. This is because early stages for generating sequential data is very inefficient using vanilla DDPMs sampling (Ho et al., 2020) as witnessed in Figure 1 (b), resulting in minor improvement of recognizability in a long period of sampling as unveiled in Figure 1 (c).

Last but not least, we demonstrate the model (without retraining) can be readily used to remedy defects in sketches due to unfaithful or incomplete drawing, by incorporating instance-aware guidance into data sampling. Motivated by recent works on guided diffusion models (Dhariwal & Nichol, 2021; Ho & Salimans, 2022), gradients of perceptual similarity (Zhang et al., 2018) between the generated data and the conditional sketch were incorporated during sampling to guide the noise prediction, thereby influencing the obtained sample at each timestep. This was done with the goal of enforcing visual similarity to the conditional, flawed sketch, while also being more appealing and recognizable after reverse-time diffusion.

Our contributions can be summarized as follows: (i) Denoising diffusion models are exploited for sketch generation in vector format. The generative model is to learn distribution over stroke points' locations, from a deformation-based denoising process which starts from noise. (ii) The quality, i.e., recognizablity, of the generated sketches is quantifiable by leveraging the knowledge of the estimated noises during sampling. This is achieved by devising an auxiliary RNN, which is trained supervised under a pre-trained image classifier, to predict the recognizability of a generated sketch at timestep $t$ from the corresponding estimated noise. (iii) A shortcut sampling path can be discovered through a simple skip strategy based on the learned quality measurement net. This allows faster and more effective generation with little trade off in data quality. (iv) Instance-aware guidance built on perceptual metric is embedded into the reverse-time diffusion. It enables our model to recover distorted or corrupted sketches without retraining.

## 2 RELATED WORKS

**Sketch Generation** There is a rich literature of research works related to sketch generation. Early works (Guo et al., 2007; Li et al., 2019) leverage edge maps as substitution for sketches. Coupled

with deep learning, much progress has been made recently. Particularly, generative adversarial networks (GANs) have motivated extensive works in sketch synthesis, involving doodle-sketch generation (Ge et al., 2020), image-to-sketch translation (Liu et al., 2020), colored sketch rendering(Rathod et al., 2021), pencil-shading sketch generation (Li et al., 2020b), and face sketch synthesis (Wang et al., 2020). However, those models are all pixel-based generation, which is fundamentally different from how humans sketch objects using pens or brushes. Towards modeling sketches like humans, sketches are preferred to be treated as sequential pen actions, and RNN-based variational autoencoder (VAE) (Ha & Eck, 2018; Zhang et al., 2017; Graves, 2013), reinforcement learning (RL) (Xie et al., 2013; Zheng et al., 2018; Ganin et al., 2018), Transformed-based sketch representation (Ribeiro et al., 2020; Lin et al., 2020), learning parametric Bézier curve (Das et al., 2020; 2021b), and neural ODE (Das et al., 2021a) are explored for sketch generation. (Aksan et al., 2020) proposes a relational model built on auto-encoder, which can decompose sketch formed by a single temporal sequence into a group of disordered strokes. Particularly, promising generative results on complex structures have been witnessed. Other notable works include generating stylized line drawing from 3D shapes (Liu et al., 2021), and intent communication through sketching by referential communication game (Mihai & Hare, 2021).

**Diffusion Models** Recently, approaches with diffusion models have delivered impressive results on several generative tasks, including image generation (Ho et al., 2020; Dhariwal & Nichol, 2021), shape generation (Cai et al., 2020), 3D shape modelling (Luo & Hu, 2021), audio synthesis (Kong et al., 2020), and cross-domain generation (Popov et al., 2021; Nichol et al., 2021). Different from likelihood-based models (variational auto-encoder (VAEs) (Kingma & Welling, 2013), normalizing flow models (Dinh et al., 2014; Papamakarios et al., 2021), energy-based models (EBMs) (LeCun et al., 2006)) and implicit generative models (GANs (Goodfellow et al., 2014)), diffusion models can be categorized into score-based generative modeling (SGM) (Song et al., 2020), which aims to model the gradient of the log probability density function by score matching (Hyvärinen & Dayan, 2005). There are two popular sub-classes in SGMs, i.e., score matching with Langevin dynamics (SMLD) (Song & Ermon, 2019) and denoising diffusion probabilistic models (DDPM) (Sohl-Dickstein et al., 2015; Ho et al., 2020). Despite attractive, rare work with diffusion models targets at handling sketches. The most relevant work to ours is Diff-HW (Luhman & Luhman, 2020) which also applies diffusion models for sequential data generation, i.e., handwriting. However, Diff-HW adopts the vanilla DDPMs and focuses on text-to-sketch translation. On contrast, we offer (model built-in) quality quantifiable diffusion models, improved sampling strategy and gradients guided conditional sampling based on DDIM.

## 3 DIFFUSION MODELS FOR VECTORIZED SKETCH GENERATION

Our objective is to generate sketch stroke sequences from noise by a novel method built on denoising diffusion implicit models (DDIMs). DDIMs generalize DDPMs by introducing a non-Markovian diffusion process, which yet still has the same forward marginals as DDPMs. Uniquely, we propose a method to find a shortcut sampling trajectory based on *recognizablity* of a generated sketch. Moreover, we will present how to rectify a flawed sketch by our trained unconditional sketch generation model on-the-fly during the generative process.

### 3.1 PROBLEM SETUP

We construct a sketch in a sequence as $s_0 = \{s^1, s^2, \ldots, s^N\}$ using the representation from Ha & Eck (2018), i.e. *stroke-3*. Each point $s^i$ is represented as a 3-D vector $(\Delta x^i, \Delta y^i, g^i)$, where $(\Delta x^i, \Delta y^i)$ indicates the offsets at point $i$ during the pen's moving trajectory, and $g^i$ is a binary pen state, denoting whether the pen is touching the paper or not. Our goal is to learn the probability distribution of the offsets $\{(\Delta x^i, \Delta y^i)\}$ from the training data by diffusion models. Then, a sketch can be drawn given the estimated $(\Delta x^i, \Delta y^i)$ for each point and the corresponding pen state $p^i$ inferred by a pen-state network. We will describe more details in the following sections.

### 3.2 SKETCH DIFFUSION IN FORWARD PROCESS

During the forward process, the noise will be gradually added to each point offsets of an original sketch, resulting in increased stroke distortion over time. Consider we have $N$ ordered stroke points

for a given sketch $s_0 = \{s^1, s^2, \ldots, s^N\}$, the offset $(\Delta x^i, \Delta y^i)$ of each point $s^i$ is supposed to be sampled independently from a distribution $q(s_0)$. To diffuse $s_0$ into $s_1, \ldots, s_T$, the Markov diffusion process introduced in DDPMs is applied here. Formally, the forward process enforces each of the point offsets $(\Delta x^i, \Delta y^i)$ in $s_0$ to drift along both $x$ and $y$ coordinates by gradually adding noises sampled from Gaussain distributions with pre-defined schedules $\alpha_1, \ldots, \alpha_T$. Therefore, the Markov chain in forward process is defined by:

$$q(s_{1:T}|s_0) := \prod_{t=1}^{T} q(s_t|s_{t-1}), \quad q(s_t|s_{t-1}) := \mathcal{N}\left(s_t; \sqrt{\frac{\alpha_t}{\alpha_{t-1}}} s_{t-1}, \left(1 - \frac{\alpha_t}{\alpha_{t-1}}\right) I\right). \quad (1)$$

### 3.3 DDIM BASED GENERATIVE PROCESS

Following (Song et al., 2021), DDIM-based sampling is adopted to generate sketches from noise, i.e., the reverse process, as it normally achieves high data quality with significantly small sampling steps. More importantly, their non-Markovian sampling process supports us in discovering a novel shortcut sampling function being more efficient and effective for our sketch generation. Formally, the generative process is defined as follows:

$$q_\sigma(s_{t-1}|s_t, s_0) = \mathcal{N}\left(\sqrt{\alpha_{t-1}} s_0 + \sqrt{1 - \alpha_{t-1} - \sigma_t^2} \cdot \frac{s_t - \sqrt{\alpha_t} s_0}{\sqrt{1 - \alpha_t}}, \sigma_t^2 I\right). \quad (2)$$

With DDIM we can predict $s_0$ through the generative process

$$p_\theta^{(t)}(s_{t-1}|s_t) = \begin{cases} \mathcal{N}(f_\theta^{(1)}(s_1), \sigma_1^2 I) & \text{if t=1} \\ q_\sigma(s_{t-1}|s_t, f_\theta^{(t)}(s_t)), & \text{otherwise} \end{cases} \quad (3)$$

where $f_\theta^{(t)}(s_t) = (s_t - \sqrt{1 - \alpha_t} \cdot \epsilon_\theta^{(t)}(s_t))/\sqrt{\alpha_t}$ is a prediction of $s_0$ based on our noise approximator $\epsilon_\theta^{(t)}$. Then we can sample a data sample $s_0$ from a random noise $s_T = \mathcal{N}(0, I)$ by iteratively repeating the following equation

$$s_{t-1} = \sqrt{\alpha_{t-1}} \left(\frac{s_t - \sqrt{1 - \alpha_t}\epsilon_\theta^{(t)}(s_t)}{\sqrt{\alpha_t}}\right) + \sqrt{1 - \alpha_{t-1} - \sigma_t^2} \cdot \epsilon_\theta^{(t)}(s_t). \quad (4)$$

More detailed derivations about DDPMs and how DDIMs generalize DDPMs by defining a non-Markovian forward process and obtain the corresponding generative process described in Eq. 3 can be found in Appendix A.1 and Appendix A.2.

### 3.4 NOISE APPROXIMATOR $\epsilon_\theta^{(t)}$

In our case the noise approximator $\epsilon_\theta^{(t)}(s_t)$ in Eq (4) is a trainable network to estimate noise $\epsilon^{(t)} \in \mathbb{R}^{N \times 2}$ as coordinate offsets for $s_t \in \mathbb{R}^{N \times 2}$ at timestep $t$. An improved U-Net is developed to handle the sequence of the points from a vectorized sketch. Namely, we add a trainable embedding and a decoding layer into U-Net to transform the input $s_t$ into an embedding $e_t \in \mathbb{R}^{N \times 128}$ and convert the penultimate feature embedding back to coordinates. Then the rest design is as per the conventional U-Net for dealing with 2D images. Please refer to Appendix A.3 for details.

### 3.5 RECOGNIZABILITY BASED SHORTCUT SAMPLING

As shown in Figure 1 (c), we observe that at the early sampling steps, the recognizability of generated $s_t$ is consistently low, although the amount of the estimated noise is large if we take full reverse steps of length $T$ used in DDPMs. In contrast, denoising gets much more efficient and effective, leading to a noticeable leap of recognizability at the later steps. We then intuitively suppose that the recognizability of a generated sample ought to be inferred from the pattern of denoising sequence for all stroke points, i.e., $\epsilon_\theta^{(t)} \in \mathbb{R}^{N \times 2}$. Therefore, we incorporate an additional trainable network to predict the recognizability $r_t$ given the estimated noise:

$$\hat{r}_t = h_\phi(\epsilon_\theta^{(t)}, t). \quad (5)$$

And the prediction $\hat{r}_t$ could be used as a signal to find a shortcut sampling path – we can skip $m$ sampling steps if $\hat{r}_t < \zeta$, which implies the period of ineffective sampling. $m$ is a constant indicating the interval steps to skip, and $\zeta$ is a predefined threshold. Therefore the timestep sampling function can be formulated as

$$t_n = \begin{cases} t_c - m, & \text{if } h_\phi(\epsilon_\theta^{(t_c)}, t_c) < \zeta, \\ t_c - 1, & \text{otherwise}, \end{cases} \tag{6}$$

where $t_c$ is the current timestep and $t_n$ is the next sampled timestep. Different from the Linear or Quadratic sub-sequence selection proposed in DDIM Song et al. (2021), our sub-sequence is chosen adaptively, which can be more effective and shorter.

In practice, $h_\phi(\cdot)$ is implemented as a bi-directional RNN as (Ha & Eck, 2018) with the estimated sequence of noise as input. Then the output latent vector $z \in \mathbb{R}^d$ is fed into a trainable linear layer to predict $\hat{r}_t$. And the ground truth $r_t$ could be obtained from an extra pre-trained sketch classifier as done in (Song et al., 2018). Namely, we use the maximum probability of the softmax prediction as $r_t$. Once trained, $h_\phi(\cdot)$ could be accommodatingly used to inspect how recognizable a sketch is during sampling, while no need to render it into an image. Please refer to Appendix A.4 for more analysis and insights about the shortcut sampling.

## 3.6 Pen State Estimation

The diffusion model is to learn the distribution of the coordinate offsets for sketch points. However, it is also required to predict binary pen state for each of the stroke point. Following (Luhman & Luhman, 2020), the feature vector, i.e., $v \in \mathbb{R}^{n \times 128}$, from the penultimate layer of the above U-Net for noise approximation is utilized. We feed this feature sequence into another trainable linear layer followed by a sigmoid function to predict each point's pen status:

$$\hat{g} = \texttt{sigmoid}(f_\psi(v)), \tag{7}$$

where $\psi$ are parameters of the trainable linear layer. When $\hat{g}^i > 0.5$, it indicates the pen is touching the canvas at point $i$. We do this for each timestep in the generative process.

## 3.7 Training Objective of $\epsilon_\theta^{(t)}$

To train our noise approximator $\epsilon_\theta^{(t)}$, we follow Luhman & Luhman (2020) to jointly minimize the denoising loss and pen state loss. Specifically, we train $\epsilon_\theta^{(t)}$ following DDIMs Song et al. (2021) to minimize the L2 discrepancy between the estimated noise $\epsilon^{(t)}$ and $\epsilon_t$ generated in the forward process:

$$L_\text{d}(\theta) = \mathbb{E}||\epsilon^{(t)} - \epsilon_\theta^{(t)}(s_t)||_2^2. \tag{8}$$

Additionally, at each timestep $t$, $\epsilon_\theta^{(t)}$ is also optimized by minimizing the pen state loss

$$L_\text{p}(\theta) = \frac{1}{N} \sum_{i=1}^N [-g^i \log(\hat{g}^i) - (1 - g^i) \log(1 - \hat{g}^i)], \tag{9}$$

where $\hat{g}$ is predicted using Eq. 7. In summary, the total training loss is:

$$L(\theta) = L_\text{d}(\theta) + \gamma L_\text{p}(\theta), \tag{10}$$

where $\gamma$ is a weight to balance two losses.

## 3.8 Conditional Generation to Rectify Bad Sketches

The above has described the details of the unconditional sketch generation. We now further present how to rectify a flawed sketch using our model trained with unconditional generation only. Inspired by the *classifier guidance* widely used in diffusion models (Dhariwal & Nichol, 2021; Ho & Salimans, 2022), a sketch generation guidance is introduced for the generative process. Different from existing works using gradients of an image classifier as guidance, a perceptual metric is employed to

generate guidance – the gradients of the log probability of perceptual distance between a generated $s_t$ and a condition $s_c$, denoted as $\nabla_{s_t} \log p(s_t, s_c)$. Then the estimated noise is refined as

$$\hat{\epsilon}^{(t)}(s_t) = \epsilon_\theta^{(t)}(s_t) + \eta \nabla_{s_t} \log p(s_t, s_c), \tag{11}$$

where $\eta$ is to control the strength of the guidance. Then the sample $\hat{s}_{t-1}$ will be generated using Eq 4 with the new noise $\hat{\epsilon}^{(t)}(s_t)$. Intuitively, we aim to guide the generation of $s_0$ by repeatedly sampling $s_t$, which has similar content as $s_c$. We follow prior work (Zhang et al., 2018) based on the L2 distance between two image features to measure their perceptual similarity, as it is effective in transferring image content. Neural line rendering (NLR) (Li et al., 2020a) is used to convert a vector sketch into an image sketch to allow the gradients being propagated to the vectorized sketch. After rasterizing sketch into its image version, a perceptual metric is thus applied as

$$p(s_t, s_c) = \sum_{l=1}^{L} \frac{1}{H_l W_l} \sum_{h,w} ||w_l \odot (F_{hw}^l(s_t) - F_{hw}^l(s_c))||_2^2, \tag{12}$$

where $F_{hw}^l(\cdot) \in \mathbb{R}^{H_l \times W_l \times C_l}$ denotes the feature maps for $l$-th layer of an ImageNet pre-trained VGG Simonyan & Zisserman (2014), $w_l \in \mathbb{R}^{C_l}$ is adopted to scale the feature activations channel-wise as per (Zhang et al., 2018). Details about NLR are provided in Appendix A.5.

## 4 EXPERIMENTS

In this section, we evaluate our model in two modes, i.e., unconditional and conditional generation, to verify the quality of the generated data and the ability to mend inferior sketches of our model. Please refer to Appendix A.6 for implementation details.

### 4.1 UNCONDITIONAL GENERATION

**Dataset.** We evaluate our proposed method on *QuickDraw* Ha & Eck (2018), which contains over 50M sketches in vector format across 345 common categories. A subset[2] from *QuickDraw* is collected for the experiments. Specifically, there are 10 classes chosen adhered to the following principles: (i) Complexity - simple, moderate and complex sketch drawings are all included, e.g., `fish`, `umbrella` and `lion`; (ii) Diversity - objects with diverse sub-category variations are involved, e.g., `bus` and `spider`; (iii) Ambiguity - sketches belong different classes share highly similar appearance, e.g., `apple` and `moon`. The original data split is adopted, i.e., 70,000 training and 2,500 testing sketches for each class.

**Competitors.** Three current RNN-based state-of-the-art methods, i.e., SketchRNN Ha & Eck (2018), SketchPix2seq Chen et al. (2017) and SketchHealer Su et al. (2020), are included. Diff-HW Luhman & Luhman (2020), which is built on DDPMs, is enabled for comparison by tweaking the cross-modal attention layers[3] to be self-attention since only sketches are available in our problem. Additionally, the unconditional mode of SketchODE (Das et al., 2021a) is also used for comparison.

**Evaluation metrics.** We gauge the quality of the generated data resorting to the evaluation metrics for image generation once the vector sketches are rasterized into images. Specifically, Fréchet Inception Distance (FID) Heusel et al. (2017) measures the distance between the generated (image) data and real ones by comparing the mean and variance of image features, which are obtained from Inception-V3 Szegedy et al. (2016) trained on ImageNet Krizhevsky et al. (2012) for image classification. The Geometry Score (GS) Khrulkov & Oseledets (2018) metric compares the geometrical properties of data manifold between the generated and real data. In addition, the improved precision and recall (Kynkäänniemi et al., 2019) are used as complementary evaluation metrics following other image generation works (Nichol & Dhariwal, 2021).

**Qualitative results.** Figure 2(a) shows some examples of reverse-time diffusion process, i.e., from random noise till reach the data sample, the generated sketch at each step exhibits different (reduced) level of distortion. More results of unconditional sketch generation are demonstrated in Figure 2(b).

---

[2]Class list: `moon`, `airplane`, `fish`, `umbrella`, `train`, `spider`, `shoe`, `apple`, `lion`, `bus`.
[3]Diff-HW is originally proposed for stylized text-to-handwriting generation, requiring text and an image as condtions to control the content and style of the generated handwriting, respectively.

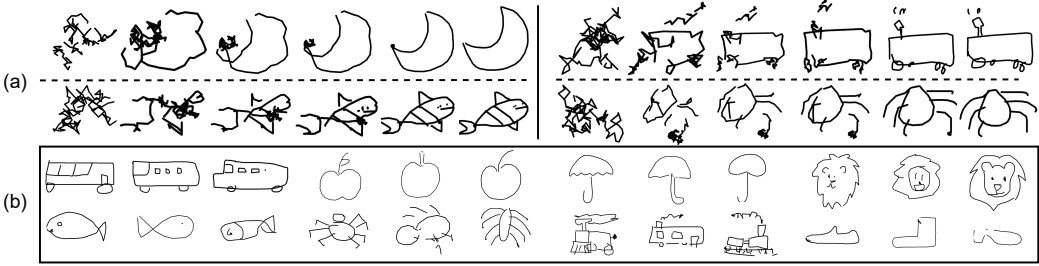

Figure 2: (a) Sketch generated from random noise. (b) More examples of unconditional generation.

Table 1: Quantitative comparison results. Testing categories are orgnaized in three folds according to the complexity, i.e., the average number of stroke points (ASP). Simple: $< 40$ ASP, Moderate: $40 \sim 100$ ASP and Complex: $> 100$ ASP. Speed: second per sketch sampling.

| Model | Simple | | | | Moderate | | | | Complex | | | | Speed↓ |
|---|---|---|---|---|---|---|---|---|---|---|---|---|---|
| | FID↓ | GS↓ | Prec↑ | Rec↑ | FID↓ | GS↓ | Prec↑ | Rec↑ | FID↓ | GS↓ | Prec↑ | Rec↑ | |
| SketchPix2seq | 13.3 | 7.0 | 0.40 | 0.79 | 16.4 | 49.7 | 0.38 | 0.75 | 18.0 | 73.3 | 0.36 | 0.72 | 0.04 |
| SketchHealer | 10.3 | 5.9 | 0.45 | 0.81 | 12.9 | 9.8 | 0.39 | 0.79 | 25.9 | 93.2 | 0.29 | 0.63 | 0.03 |
| SketchRNN | 10.8 | 5.4 | 0.44 | 0.82 | 13.0 | 11.0 | 0.42 | 0.77 | 21.4 | 97.6 | 0.35 | 0.72 | 0.03 |
| Diff-HW | 13.3 | 6.8 | 0.42 | 0.81 | 15.9 | 23.4 | 0.37 | 0.76 | 18.3 | 64.4 | 0.23 | 0.64 | 0.19 |
| SketchODE | 11.5 | 9.4 | 0.48 | 0.74 | 18.8 | 29.6 | 0.31 | 0.66 | 33.5 | 68.1 | 0.20 | 0.58 | 0.03 |
| Ours (full 1000 steps) | **6.9** | **3.4** | **0.52** | **0.88** | **8.4** | **4.7** | **0.45** | **0.87** | **9.4** | **5.2** | **0.42** | **0.85** | 1.29 |
| Ours ($r$-Shortcut, S=30) | 7.4 | 3.9 | 0.47 | 0.87 | 8.9 | 5.2 | 0.44 | 0.85 | 10.5 | 6.1 | 0.39 | 0.81 | 0.08 |
| Ours (Linear-DDIMs, S=30) | 11.9 | 6.4 | 0.38 | 0.81 | 13.3 | 8.8 | 0.36 | 0.78 | 15.1 | 9.6 | 0.33 | 0.72 | 0.08 |
| Ours (Quadratic-DDIMs, S=30) | 12.3 | 6.6 | 0.41 | 0.79 | 13.8 | 8.7 | 0.35 | 0.76 | 15.4 | 9.9 | 0.34 | 0.75 | 0.09 |
| Ours (Abs) | 20.7 | 12.1 | 0.18 | 0.55 | 23.4 | 64.6 | 0.13 | 0.48 | 29.4 | 98.9 | 0.10 | 0.39 | 0.20 |
| Ours (Point-Shuffle) | 9.5 | 5.3 | 0.35 | 0.72 | 11.3 | 7.5 | 0.31 | 0.65 | 12.4 | 8.1 | 0.20 | 0.61 | 0.18 |
| Ours (Stroke-Shuffle) | 8.2 | 3.8 | 0.36 | 0.74 | 9.6 | 7.4 | 0.34 | 0.66 | 10.3 | 7.6 | 0.25 | 0.62 | 0.18 |

**Quantitative results.** As shown in Table 1, our sketch generator clearly outperforms other competitors, suggesting better quality of the generated data. Particularly, our model is able to maintain relative stable FID, GS, Precision and Recall scores regardless of the complexity of the generated sketches. On the contrary, obvious performance decline is witnessed when constructing sketches with more complicated structures for other baseline methods.

**Sampling trajectory .** We compare different approaches for choosing sampling trajectory, including full reverse steps, linear and quadratic in DDIMs and ours using recognizability-based skip function. Results in Table 1 show that our recognizability-based skipping ($r$-Shortcut) achieves the best results when performing the same total sampling steps, i.e., $S = 30$, suggesting the superiority of our sampling strategy. In addition, compressing sampling steps from 1000 to 30 using our method deteriorates the data quality very minor, while 16x faster speed is reached.

**Effectiveness of $h_\phi(\cdot)$.** We further conduct experiments to testify if the learned network $h_\phi(\cdot)$ in Eq.5 can faithfully reflect the recognizability of sketches generated during sampling. Specifically, we compare how well the predicted recognizability can match the probability of assigning the correct class label given by the pre-trained classifier. This experiments are conducted using 10 single class models, thus we can know which class probability in the classifier to be compared. Results in Table 2 show that low and stable error can be achieved along the sampling steps.

**Impact of $N$.** To study the impact of point number $N$, we train our model with different settings. We can see from Table 3 that it is inferior to using too fewer points ($N = 24$) for modeling relative complex structures, while applying too much points ($N = 384$) is also sub-optimal. An unique and optimal $N$ is hard to reach as the complexity of sketch structure varies case by case.

**Absolute coordinates works?** To gain more insights about the importance of modeling relative coordinates by our model, i.e., $(\Delta \boldsymbol{x}, \Delta \mathbf{y})$, we train a model to learn from sketches represented by stroke points in absolute coordinates, i.e., $(\boldsymbol{x}, \mathbf{y})$, with other settings/components unchanged, denoted as "Ours (Abs)" in Table 1. Significant decrease on performances is observed, verifying the crucial role of training model with relative coordinates.

**What is learned?** We suspect that capturing the implicit drawing structure is the key to success. A simple way to verify the speculation could be destroying the drawing structure. Specifically, we reorganize the original sketch data into a disordered version by randomly shuffling sketch segments

Table 2: Averaged error (10k samples for each class) of the predicted recognizability using $h_\phi(\cdot)$.

| t | 10 | 30 | 50 | 80 | 100 | 200 | 300 | 400 | 500 | 600 | 700 | 800 | 1000 |
|---|---|---|---|---|---|---|---|---|---|---|---|---|---|
| Error | 0.0608 | 0.0619 | 0.0723 | 0.0877 | 0.1006 | 0.0928 | 0.0935 | 0.0876 | 0.0891 | 0.0819 | 0.0908 | 0.0972 | 0.1027 |

Table 3: Impact of points number $n$ (full reverse steps are performed).

| | | Simple | | | | Moderate | | | | Complex | | | | Speed |
|---|---|---|---|---|---|---|---|---|---|---|---|---|---|---|
| | | FID | GS | Prec | Rec | FID | GS | Prec | Rec | FID | GS | Prec | Rec | |
| | N=24 | 7.4 | 4.4 | 0.32 | 0.74 | 14.7 | 13.5 | 0.30 | 0.76 | 16.3 | 18.2 | 0.26 | 0.69 | 0.16 |
| T=1000 | N=192 | 6.9 | 3.4 | 0.33 | 0.76 | 9.8 | 5.3 | 0.31 | 0.74 | 10.3 | 6.3 | 0.27 | 0.71 | 0.18 |
| | N=384 | 8.5 | 5.0 | 0.30 | 0.72 | 10.6 | 8.1 | 0.28 | 0.69 | 11.9 | 12.6 | 0.23 | 0.67 | 0.22 |

(each sketch segment is formed by connecting two neighbor stroke points). Intuitively, the newly constructed sequential data discards the implicit interdependent relations among stroke points, as the corresponding segments are geometrically far from each other. Then we can obtain a variant model trained with such structure-broken data, denoted by "Ours (Point-Shuffle)" in Table 1. We can see that the performance is clearly harmed compared to the model trained with normal data. Furthermore, we also experiment with a model variant trained with a stroke-level shuffle tactic, which means that the structural cues at *stroke-level* are preserved as each stroke remain complete, however the stroke orders are changed given the random shuffle applied. Less declines are observed ("Ours (Stroke-Shuffle)" in Table 1), revealing that the middle-level structural cues (i.e., complete strokes) are of great importance to our model. Note full steps are taken for these two model variants.

## 4.2 CONDITIONAL GENERATION

Experiments on sketch refinement and sketch healing are conducted to verify the effectiveness of the conditional sampling by our model. We show that, given any conditional sketch with defeats due to either stroke distortion or corruption, the generated sketch could be a refined version accordingly.

**Dataset.** We utilize the same data in sec 4.1 for the sketch refinement task. To synthesis sketches with different degrees of distortion, random Gaussian noise $\mathbf{e} \sim \mathcal{N}(0, \boldsymbol{I})$ is added to each location of stroke points. The deformation degrees could be easily controlled by adding noise using different $t$ timesteps. Because a real sketch can be transformed into a random noise after $T$ diffusion steps, then a new sample obtained from the intermediate step $t$ would be considered as $x\%$ deformed if $t = x\%T$. The sketch healing task aims to create new sketches which should resemble the given the partial sketches. For fair comparison, we follow the same experimental setups in SketchHealer (Su et al., 2020). 17 categories from *QuickDraw* and the same data splits (70,000 for training and another 2,500 novel ones for testing per class) are adopted. Sketches are damaged using two different mask ratios: $p_{mask} = 10\%$ or $30\%$, that is, $10\%$ or $30\%$ key stroke points will be randomly removed to form the corrupted versions from a complete sketch.

**Sketch classifier.** A multi-category classifier built on AlexNet is pre-trained on the training set of all 345 *QuickDraw* categories. Intuitively, better recognizability of sketches is manifested by higher recognition accuracy offered by this pre-trained classifier. Following the common practice in Song et al. (2018); Su et al. (2020), acc@1 and acc@10 are adopted as evaluation metrics. Specifically, acc@K denotes the accuracy of the true positive is ranked within the top K predictions.

**SBIR model.** Similarly, a sketch-based image retrieval (SBIR) model is pre-trained to verify if the rectified sketch generated by our model could enable better retrieval performances. Specifically, Triplet-SAN Yu et al. (2016), which is constructed by employing Sketch-A-Net Yu et al. (2017) as backbone network, is trained on the QuickDraw Extended dataset Dey et al. (2019) under the supervision of triplet loss. Built on *QuickDraw*, QuickDraw Extended dataset is the largest SBIR dataset which contains 330,000 sketches paired with 204,000 photos over 110 categories. In our case, the SBIR model is trained by using the training set of the same selected 10 categories in the section 4.1. The remaining testing set of the 10 classes is utilized as samples to be deformed. Evaluation metric acc@1 and acc@10 measure if a target image could be ranked within the top 1/10 retrieval results. The mean average precision (mAP) is adopted for evaluation as well.

**Results**. As shown in Table 4, we can observe that (i) for all cases involving noisy sketches (i.e., deformation levels from $10\% \sim 50\%$), performance improvements on recognition and retrieval can be achieved by using sketches after rectification, revealing the obvious enhancement benefits gained from our model; (ii) The recognition and retrieval results are stable regardless of the deformation level tackled, e.g., recognition accuracy acc@1 is kept around $47\% \sim 50\%$ for sketches

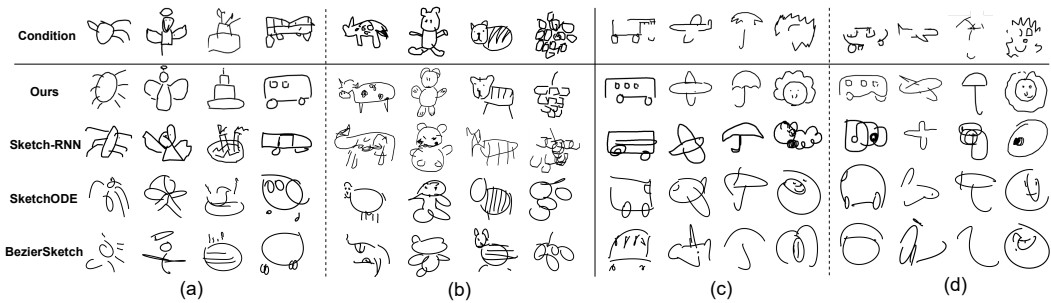

Figure 3: Qualitative comparison on sketch refinement (a)(b) and healing (c)(d). (a) Normal degree of complexity. (b) Complex classes. (c) Corruption $p_{mask} = 10\%$. (d) Corruption $p_{mask} = 30\%$.

Table 4: Recognition and retrieval results before and after (separated by "|") sketch rectification by our model at different deformation levels (DL). Performance gain in red. "–": without injecting noise, i.e., real data.

| DL | Recognition | | | | Retrieval | | | | |
|----|-------------|--|--|--|-----------|--|--|--|--|
| | acc@1(%) | | acc@10(%) | | mAP | | acc@1(%) | | acc@10(%) | |
| – | 51.9 | 52.4 (+0.90) | 87.7 | 90.2 (+2.50) | 0.704 | 0.789 (+0.045) | 67.4 | 73.3 (+5.90) | 91.3 | 96.2 (+4.90) |
| 10% | 45.7 | 48.9 (+3.20) | 82.3 | 82.4 (+0.10) | 0.724 | 0.788 (+0.064) | 66.9 | 73.1 (+6.20) | 92.1 | 96.8 (+4.70) |
| 20% | 33.0 | 47.3 (+14.3) | 68.2 | 81.9 (+13.7) | 0.607 | 0.772 (+0.165) | 55.8 | 72.8 (+17.0) | 81.8 | 94.7 (+12.9) |
| 30% | 20.6 | 48.2 (+27.6) | 51.5 | 81.9 (+30.4) | 0.496 | 0.787 (+0.291) | 46.9 | 72.8 (+25.9) | 68.9 | 95.0 (+26.1) |
| 50% | 7.29 | 50.1 (+42.8) | 27.1 | 84.3 (+57.2) | 0.328 | 0.786 (+0.458) | 28.6 | 74.9 (+46.3) | 47.8 | 96.3 (+48.5) |

with deformation $10\% \sim 50\%$. (iii) Interestingly, recognition and SBIR results could be further significantly increased on the original human-drawn sketches after refinement. Exemplar refined sketches from the original ones are demonstrate in Figure 3 (a), revealing that badly drawn sketches (i.e., missing part, random strokes and line distortions) can be largely rectified by our model, while other competitors either not able to faithfully resemble the conditions or unable to model complex structures. Quantitative healing results are shown in Table 5. We can see that our generator outperforms all the other competitors on sketch recognition in most cases (expect the top 10 result when $p_{mask} = 10\%$), indicating the superiority of our model on recovering incomplete sketches. The advantage of our method is further enlarged when the corruption level increases, i.e., $p_{mask} = 30\%$. Similar situation can be observed for human study results. The healed sketches given by our model are mostly preferred compared against with other baseline methods. Qualitative comparisons against Sketch-RNN, SketchODE and BézierSketch are also provided in Figure 3 (b).

Table 5: Comparison results on sketch healing. Recognition results are obtained by classifying generated healed sketches with a pre-trained multi-category sketch classifier. "Human" denotes human's preference of choice among the synthetic outputs by different competitors.

| $p_{mask}$ | Metric | | SketchRNN | SketchPix2seq | SketchHealer | Ours |
|---------|--------|--------|-----------|---------------|--------------|------|
| 10% | Recognition | Top 1 | 24.41% | 21.88% | 49.77% | **56.61%** |
| | | Top 5 | 46.23% | 31.92% | 69.92% | **71.63%** |
| | | Top 10 | 56.28% | 36.91% | **80.01%** | 79.79% |
| | Human | N/A | 21.11% | 10.82% | 33.56% | **34.51%** |
| 30% | Recognition | Top 1 | 3.14% | 9.51% | 41.59% | **55.88%** |
| | | Top 5 | 10.25% | 16.06% | 62.76% | **74.57%** |
| | | Top 10 | 15.91% | 20.26% | 68.12% | **80.70%** |
| | Human | N/A | 6.14% | 6.98% | 30.76% | **56.12%** |

## 5 CONCLUSION

We show for the first time sketch generation can be formulated as a process of deformation-based denoising. The key finding is that increased sketch deformation degrees can be monotonically synthesized by diffusing stroke points with Gaussian noise, and the demanded probabilistic distribution of the stroke points of sketch objects can thus be effectively learned by diffusion inversion. Importantly, the ability of quantifying recognizability of the generated sketch was injected during the sampling. For that, a RNN was developed to predict the recognizability of a sampled sketch based on the estimated noise at each timestep. As a result, more efficient sampling can be enabled by a recognizability-based skip function. Additionally, our model trained for unconditional generation could be readily extended for conditional generation by incorporating a perceptual similarity based gradients into the sampling. Extensive experiments validated the effectiveness of our model. To manage the abstraction level of the generated sketch would be a potential future work.

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

# A APPENDIX

## A.1 DENOISING DIFFUSION PROBABILISTIC MODELS (DDPM)

To learn the probability distribution over data $\mathbf{x}$, diffusion models corrupt training data by slowly injecting noise and learn to reverse the corruption, such that the obtained models can gradually transform random noise into sample for data generation.

**Forward process.** Formally, for each training data $\mathbf{x}_0 \sim q_{data}(\mathbf{x}_0)$, a discrete Markov chain $\mathbf{x}_0, \mathbf{x}_1, \ldots, \mathbf{x}_T$ is formed by the forward process (also known as diffusion process). This process is defined as a Markov chain which slowly adds Gaussian noise to the data according to a variance schedule $\beta_1, \ldots, \beta_T$:

$$q(\mathbf{x}_{1:T}|\mathbf{x}_0) := \prod_{t=1}^{T} q(\mathbf{x}_t|\mathbf{x}_{t-1}) \tag{13}$$

$$q(\mathbf{x}_t|\mathbf{x}_{t-1}) := \mathcal{N}(\mathbf{x}_t; \sqrt{1-\beta_t}\mathbf{x}_{t-1}, \beta_t\mathbf{I}) \tag{14}$$

If we know $q(\mathbf{x}_{t-1}|\mathbf{x}_t)$, we could sample data from the data distribution $q(\mathbf{x}_0)$ by first sampling $\mathbf{x}_T$ from $q(\mathbf{x}_T)$ (isotropic Gaussian) and then sampling from $q(\mathbf{x}_{t-1}|\mathbf{x}_t)$ until we get $\mathbf{x}_0$. However, it is difficult to estimate $q(\mathbf{x}_{t-1}|\mathbf{x}_t)$ since it needs entire dataset to do so. Therefore, $p_\theta$ is proposed to approximate the conditional probabilities $q(\mathbf{x}_{t-1}|\mathbf{x}_t)$ during the backward process.

**Backward process.** In the backward/reverse process, diffusion models have to denoise the perturbed data (starting at random noise $p(\mathbf{x}_T) = \mathcal{N}(\mathbf{x}_T; \mathbf{0}, \mathbf{I})$) back to the origin data $\mathbf{x}_0$. Mathematically, diffusion models is defined as

$$p_\theta(\mathbf{x}_0) := \int p_\theta(\mathbf{x}_{0:T}) d\mathbf{x}_{1:T} \tag{15}$$

in which the joint distribution $p_\theta(\mathbf{x}_{0:T})$ defines the reverse process:

$$p_\theta(\mathbf{x}_{0:T}) := p(\mathbf{x}_T) \prod_{t=1}^{T} p_\theta(\mathbf{x}_{t-1}|\mathbf{x}_t) \tag{16}$$

$$p_\theta(\mathbf{x}_{t-1}|\mathbf{x}_t) := \mathcal{N}(\mathbf{x}_{t-1}; \mu_\theta(\mathbf{x}_t, t), \mathbf{\Sigma}_\theta(\mathbf{x}_t, t)) \tag{17}$$

The training objective is to optimize variational bound on negative log likelihood:

$$\mathbb{E}[-\log p_\theta(\mathbf{x}_0)] \leq \mathbb{E}_q[-\log \frac{p_\theta(\mathbf{x}_{0:T})}{q(\mathbf{x}_{1:T}|\mathbf{x}_0)}]$$
$$= \mathbb{E}_q[-\log p(\mathbf{x}_T) - \sum_{t\geq 1} \log \frac{p_\theta(\mathbf{x}_{t-1}|\mathbf{x}_t)}{q(\mathbf{x}_t|\mathbf{x}_{t-1})}] \tag{18}$$

which is equivalent to optimize the following variational lower-bound $L_{vlb}$:

$$L_{vlb} := L_0 + L_1 + \cdots + L_{T-1} + L_T \tag{19}$$

$$L_0 := -\log p_\theta(\mathbf{x}_0|\mathbf{x}_1) \tag{20}$$

$$L_{t-1} := D_{KL}(q(\mathbf{x}_{t-1}|\mathbf{x}_t, \mathbf{x}_0)||p_\theta(\mathbf{x}_{t-1}|\mathbf{x}_t)) \tag{21}$$

$$L_T := D_{KL}(q(\mathbf{x}_T|\mathbf{x}_0)||p(\mathbf{x}_T)) \tag{22}$$

Essentially, the above KL terms compare two Gaussian distributions which can be addressed in closed form Ho et al. (2020). The training objective for Eq (17) is to get $\mu_\theta(\mathbf{x}_t, t)$, while not

involve $\boldsymbol{\Sigma}_\theta(\mathbf{x}_t, t)$, as it is set to time-dependent constants $\sigma_{t=1}^2\mathbf{I}$. Furthermore, instead of predicting $\mu_\theta(\mathbf{x}_t, t)$ (forward process posterior mean) by a neural network, Ho et al. (2020) proposed to utilize an approximator $\epsilon_\theta(\mathbf{x}_t, t)$ to predict noise $\epsilon$ from $\mathbf{x}_t$, which is proven to be more effective than optimizing $\mu_\theta(\mathbf{x}_t, t)$. The simplified training objective is:

$$L_{simple}(\theta) := \mathbb{E}_{t\sim[1,T],\mathbf{x}_0\sim q(\mathbf{x}_0),\epsilon\sim\mathcal{N}(0,\mathbf{I})}[||\epsilon - \epsilon_\theta(\mathbf{x}_t, t)||^2] \tag{23}$$

**Data sampling.** Once trained, we have a neural network to estimate noise $\epsilon$ from sample $\mathbf{x}_t$ at timestep $t$, i.e., $\epsilon_\theta(\mathbf{x}_t, t)$. Then $\mu_\theta(\mathbf{x}_t, t)$ can be derived from $\epsilon_\theta(\mathbf{x}_t, t)$ by the following equation:

$$\mu_\theta(\mathbf{x}_t, t) = \frac{1}{\sqrt{\alpha_t}}(\mathbf{x}_t - \frac{1-\alpha_t}{\sqrt{1-\bar{\alpha}_t}}\epsilon_\theta(\mathbf{x}_t, t)) \tag{24}$$

where $\alpha_t = 1-\beta_t$ and $\bar{\alpha}_t := \prod_{s=1}^t \alpha_s$. To this end, we can sample data from $p_\theta(x_{t-1}|x_t)$ according to Eq (17) repeatedly until we reach $\mathbf{s}_0$.

## A.2 DENOISING DIFFUSION IMPLICIT MODELS (DDIMs)

To improve the sampling efficiency, authors of DDIM (Song et al., 2021) have proposed a novel non-Markov chain process to reduce the forward and reverse process steps of DDPMs. They found a special property of the forward process of DDPM as

$$q(\boldsymbol{x}_t|\boldsymbol{x}_0) = \mathcal{N}(\boldsymbol{x}_t; \sqrt{\alpha_t}\boldsymbol{x}_0, (1 - \alpha_t)\boldsymbol{I}) \tag{25}$$

such that the training objective of DDPM is able to not directly based on the joint $q(\boldsymbol{x}_{1:T}|\boldsymbol{x}_0)$. Then, they derive that

$$q(\boldsymbol{x}_{t-1}|\boldsymbol{x}_t, \boldsymbol{x}_0) = \mathcal{N}\left(\sqrt{\alpha_{t-1}}\boldsymbol{x}_0 + \sqrt{1 - \alpha_{t-1} - \sigma_t^2}\cdot\frac{\boldsymbol{x}_t - \sqrt{\alpha_t}\boldsymbol{x}_0}{\sqrt{1-\alpha_t}}, \sigma_t^2\boldsymbol{I}\right), \tag{26}$$

where $t > 1$ and $q(\boldsymbol{x}_T|\boldsymbol{x}_0) = \mathcal{N}(\sqrt{\alpha_T}\boldsymbol{x}_0, (1 - \alpha_T)\boldsymbol{I})$. And based on Bayes' rule, the forward process can be derived as

$$q_\sigma(\boldsymbol{x}_t|\boldsymbol{x}_{t-1}, \boldsymbol{x}_0) = \frac{q_\sigma(\boldsymbol{x}_{t-1}|\boldsymbol{x}_t, \boldsymbol{x}_0)q_\sigma(\boldsymbol{x}_t|\boldsymbol{x}_0)}{q_\sigma(\boldsymbol{x}_{t-1}|\boldsymbol{x}_0)}, \tag{27}$$

i.e. the forward process is no longer Markovian as each $\boldsymbol{x}_t$ is dependent on $\boldsymbol{x}_{t-1}$ and $\boldsymbol{x}_0$.

During the generative process, they train a noise approximator $\epsilon_\theta^{(t)}$ to estimate $p_\theta^{(t)}(\boldsymbol{x}_{t-1}|\boldsymbol{x}_t)$ based on the probability $q_\sigma(\boldsymbol{x}_{t-1}|\boldsymbol{x}_t, \boldsymbol{x}_0)$ as follows

$$p_\theta^{(t)}(\boldsymbol{x}_{t-1}|\boldsymbol{x}_t) = \begin{cases} \mathcal{N}(f_\theta^{(1)}(\boldsymbol{x}_1), \sigma_1^2\boldsymbol{I}) & \text{if t=1} \\ q(\boldsymbol{x}_{t-1}|\boldsymbol{x}_t, f_\theta^{(t)}(\boldsymbol{x}_t)), & \text{otherwise,} \end{cases} \tag{28}$$

where $f_\theta^{(t)}(\boldsymbol{x}_t)$ is the prediction of $\boldsymbol{x}_0$ using the noise approximator $\epsilon_\theta^{(t)}$ give $\boldsymbol{x}_t$, as

$$f_\theta^{(t)}(\boldsymbol{x}_t) := (\boldsymbol{x}_t - \sqrt{1 - \alpha_t}\epsilon_\theta^{(t)}(\boldsymbol{x}_t))/\sqrt{\alpha_t}. \tag{29}$$

Therefore as they derived their generative training objective is

$$J_\sigma(\epsilon_\theta) := \sum_{t-1}^T \mathbb{E}_{\boldsymbol{x}_0\sim q(\boldsymbol{x}_0)}[||\epsilon_\theta^{(t)}(\sqrt{\alpha_t}\boldsymbol{x}_0 + \sqrt{1 - \alpha_t}\epsilon_t) - \epsilon_t||_2^2] \tag{30}$$

where $\epsilon_t \sim \mathcal{N}(\boldsymbol{0}, \boldsymbol{I})$. Once the noise approximator is trained, one can generate a sample $\boldsymbol{x}_{t-1}$ give a sample from $\boldsymbol{x}_t$ as

$$\boldsymbol{x}_{t-1} = \sqrt{\alpha_{t-1}}\left(\frac{\boldsymbol{x}_t - \sqrt{1 - \alpha_t}\epsilon_\theta^{(t)}(\boldsymbol{x}_t)}{\sqrt{\alpha_t}}\right) + \sqrt{1 - \alpha_{t-1} - \sigma_t^2}\cdot\epsilon_\theta^{(t)}(\boldsymbol{x}_t) \tag{31}$$

**Accelerated Generation Process** The DDIM authors proposed to use non-Markovian chain for the forward procedure as $q_\sigma(\boldsymbol{x}_t|\boldsymbol{x}_0)$ can be directly estimated. Therefore, they propose to use subsequence $\tau$ from $[1, \cdots, T]$ to speed up the generative process.

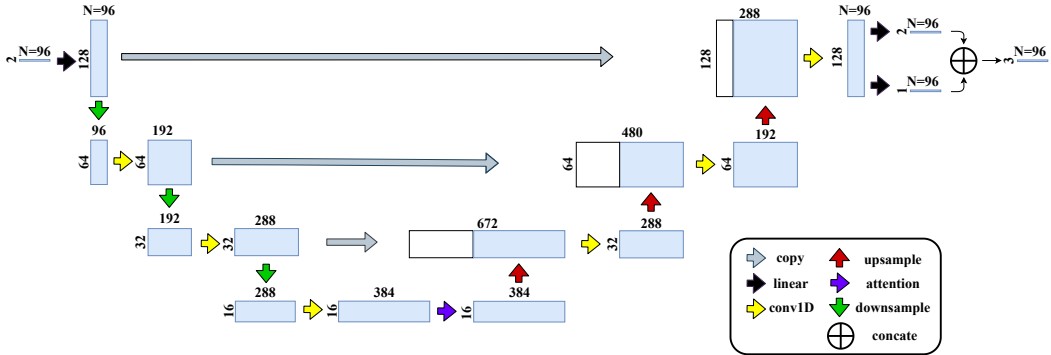

Figure 4: Architecture of the U-Net for estimating noise $\epsilon_\theta^{(t)}$ and pen state $\hat{g}$.

Table 6: Comparison results of using U-Net and Bi-directional RNN as noise approximator.

| Model | Simple | | | | Moderate | | | | Complex | | | | Speed↓ |
|---|---|---|---|---|---|---|---|---|---|---|---|---|---|
| | FID↓ | GS↓ | Prec↑ | Rec↑ | FID↓ | GS↓ | Prec↑ | Rec↑ | FID↓ | GS↓ | Prec↑ | Rec↑ | |
| Ours-UNet | 6.9 | 3.4 | 0.52 | 0.88 | 8.4 | 4.7 | 0.45 | 0.87 | 9.4 | 5.2 | 0.42 | 0.85 | 1.29 |
| Ours-biRNN | 12.3 | 5.6 | 0.42 | 0.78 | 14.7 | 18.2 | 0.39 | 0.74 | 22.8 | 86.6 | 0.24 | 0.56 | - |

## A.3 U-NET ARCHITECTURE

The architecture of our U-Net is shown in Figure 4. It consists of a stack of convolution blocks and average pooling for downsampling, followed by a stack of upsampling convolutions and convolution blocks, with skip connections for concatenating feature maps in the same resolution. Additionally, a single head attention layer is used to inject the timestep $t$ embedding into the convolution blocks during downsampling following DDIM Song et al. (2021).

*Bi-directional RNN as noise approximator.* Instead of using the conventional convolutional U-Net, a different network architecture, i.e., bi-directional RNN, is explored to estimate the noise. Specifically, given a sketch represented by a sequence of points $s_t = \{s_t^1, s_t^2, \ldots, s_t^N\}$ where $s_t \in \mathbb{R}^{N \times 2}$, bi-directional RNN is first applied to map each point $s_t^i \in \mathbb{R}^2$ into a hidden state $h_t^i \in \mathbb{R}^d$. Then all hidden states are concatenated together to construct the overall feature $h_t \in \mathbb{R}^{N \times d}$ for a sketch. Finally, $h_t$ is fed into a fully-connected layer to predict the noises, i.e., $\epsilon^t \in \mathbb{R}^{N \times 2}$. Similarly, the pen states can be obtained from $h_t$ by another FC-layer. The results are shown in Table 6. We can find out that the obtained performances can not surpass the results using U-Net.

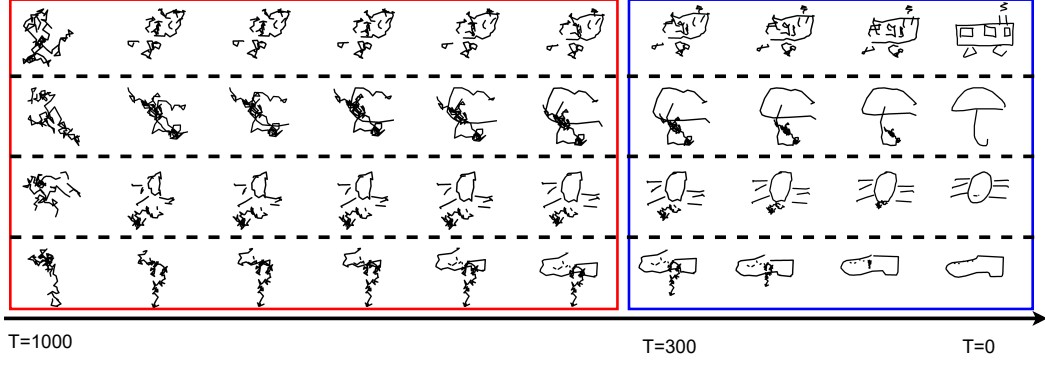

T=1000        T=300        T=0

Figure 5: Examples of the process of data sampling (one example in a row). We can see that early steps (T=1000 to about T=300) are often not effective that samples in the red box change slowly. In contrast, later steps get much more efficient to rectify sketches as in the blue box.

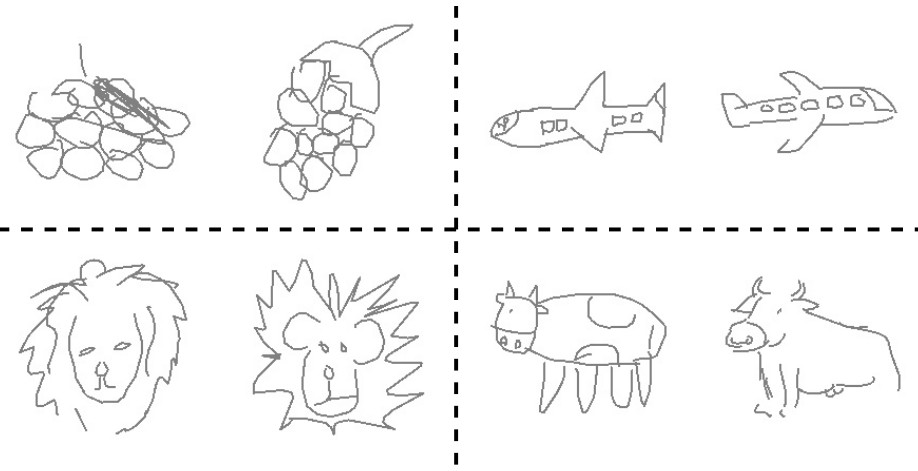

Figure 6: The obtained rasterized sketches using neural line rendering (NLR).

## A.4 FURTHER INSIGHTS INTO SHORTCUT SAMPLING

Shortcut sampling is a cute discovery we made that is specific to sketch data – coordinates in sketch sequence are far less robust to added noise than pixel values of an image. This can be seen from Figure 1(c) and Figure 5 – that early sampling steps are often long and inefficient. Addressing this merely means a speed-up in generation while retaining quality (16x faster, see Table 1).

## A.5 NEURAL LINE RENDERING

Following Li et al. (2020a), neural line rendering (NLR) is performed to convert a vectorized sketch to its pixelative image. Specifically, given a sketch represented by a sequence of points $s_t = \{s_t^1, s_t^2, \ldots, s_t^N\}$ at timestep $t$, a bi-directional Long Short-Term Memory (LSTM) is firstly used to extract per-point features $f_t^i \in \mathbb{R}^d$ for each point $s_t^i$:

$$
\begin{aligned}
[h_t^i; c_t^i] &= \text{LSTM}(s_t^i, [h_t^{i-1}; c_t^{i-1}]) \\
f_t^i &= \texttt{sigmoid}(w h_t^i + b)
\end{aligned}
\tag{32}
$$

where $h$ and $c$ are the hidden states and the optional cell states, $w$ and $b$ are the weights and biases of a fully connected layer. Then the point sequence and the features, i.e., $\{(s_t^i, f_t^i)\}$, are fed into the NLR module to produce a $d$-channel feature map of size $H \times W \times d$. And the $c$-th channel of the feature map $I^c$ (the timestep $t$ and the index for point $i$ are omitted for clarity) can be obtained as follows:

$$
I_k^c = \begin{cases} (1 - \alpha_k) \cdot f_i^c + \alpha_k \cdot f_{i+1}^c & \text{if} \quad D(I_k, p_i p_{i+1}) < \gamma \\ 0, & \text{otherwise,} \end{cases}
\tag{33}
$$

which means the pixel value $I_k^c$ is computed by a linear interpolation of $f_i^c$ and $f_{i+1}^c$ (i.e., the $c$-th feature values of two nearby stroke points $p_i$ and $p_{i+1}$) if $I_k$ is a stroke pixel (the distance from $I_k$ to the line segment $p_i p_{i+1}$ is smaller than a threshold, i.e., $D(I_k, p_i p_{i+1}) < \gamma$.) Note that $p_i$ and $p_{i+1}$ are absolute coordinates corresponding to points $s^i$ and $s^{i+1}$. And $\alpha_k = \|p^k - p_i\|_2 / \|p_{i+1} - p_i\|_2$, where $p^k$ is the projection point of $I_k$ on line segment $p_i p_{i+1}$.

To this end, by rendering the stroke points' features into pixel values $I_k$, a vectorized sketch can be transformed into an image sketch. NLR is differentiable due to the linear interpolation based rendering, thus the gradient w.r.t the perceptual similarity given by a 2D-CNN in Eq 12 can be back-propagated to the vectorized sketch. Some examples of rendered sketch images are demonstrated in Figure 6 where $d = 3$, $\gamma = 1$, $H$ and $W$ are both set to 256.

## A.6 Implementation Details

A single Nvidia 3090 GPU is used for model training. The batch size is set to 512. The point number is selected as 96, which is the average value of stroke points in the QuickDraw dataset. Cosequently, sketches with more than 96 stroke points are excluded during model training. In addition, we will pad the point sequence with zeros to reach N=96 if the actual number of stroke points of any sketch is less than 96. It turns out that $N = 96$ works well for most cases since about $91\%$ sketches (statistics from all the 345 categories) having stroke points less than 96. The default setting of skipping stride $m = 50$, and the recognizability threshold $\zeta = 0.2$. Instead of directly using $\alpha_1, \ldots, \alpha_T, \beta_1, \ldots, \beta_T$ is adopted to define the mean and variance of Gaussian noise. And a linear noise schedule is leveraged and $\beta_t$ is defined as:

$$\beta_t = \beta_1 + \frac{t-1}{T-1} \times (\beta_T - \beta_1) \tag{34}$$

where $\beta_1 = 10^{-4}$ and $\beta_T = 0.02$ in our case. Then the mean is $\sqrt{1 - \beta_t}$ and variance is $\beta_t$ in Eq 1. Adam optimizer ($\beta_1 = 0.9$ and $\beta_2 = 0.98$) is used for optimization.

