# OpenReview forum: "SketchKnitter: Vectorized Sketch Generation with Diffusion Models"
_ICLR.cc/2023/Conference — ICLR 2023 notable top 25%_

### Official Review · Reviewer_4TYn · 2022-10-24

**Confidence:** 4
**Correctness:** 4
**Technical Novelty And Significance:** 2
**Empirical Novelty And Significance:** 2
**Recommendation:** 6

**Clarity, Quality, Novelty And Reproducibility:**

Some part of the paper is quite well written, while some part can be somewhat difficult to understand. For example, the intro is quite convincing while related work can be somewhat confusing.

With that said, overall, I do think the paper is easy to understand.

The reproducibility is a bit tricky,  I don't think there is enough information to reproduce exactly the model that was used. The release of training code and pre-trained model would be required for the results to be reproducible.

**Strength And Weaknesses:**

Strength

- the method and choice of coordinate makes sense for sketch

- the results support the claim including how keeping the order is beneficial

- recognition based step skipping and rectifying bad sketches are quite interesting

- the evaluation seems quite extensive

Weakness

 - I think there are a lot of similarities with existing works such as [Luhman and Luhman, 2020] down to the use of diffusion model on stroke-3 coordinate (delta x, delta y, pen state). This means the actual novel contribution of the paper is pretty small. The paper is more about applying existing methods toward a new problem than coming up with an entirely new approach,

**Summary Of The Paper:**

The authors propose to formulate sketch generation as the reversal process of sketch deformation, allowing the use of diffusion process that's more appropriate to stroke based sketch drawing. This is quite similar to [Luhman and Luhman, 2020] which use the same approach but on handwriting generation. The authors also propose recognition guided diffusion to speed up the diffusion process, and experiment on sketch completion based on perceptual metric.

**Summary Of The Review:**

The paper is pretty much a borderline for me. On one hand, it's the first sketch generation that formulate as diffusion process, along with other nice properties, on the other hand there are quite a lot of similarities with previous works that I'm not sure if the current contributions is enough. The choice of model is also somewhat confusing to me; convolutional U-Net with slight modification (mapping to/from 128 embedding) on coordinate type of input/output (delta x, delta y, pen state). RNN, however, is used to learn recognizability of sketch. It just seems a bit odd, I would love some clarification on the motivation behind these choice of architecture.

Overall, I do think the novelty is not everything. I am somewhat positive about the key ideas of the paper; diffusion for sketch generation, formulate as iteratively deform/de-deform the sketch, recognizability to guide faster steps diffusion, rectifying bad sketch, etc. The evaluation is also quite extensive, with ablation that make sense (love the random stroke/point order). So my initial recommendation is a slightly more toward accept, as long as the authors can provide clarification to my concerns. But I can be convinced either way.

Others:
the reference for Diff-HW seems to be wrong? isn't it out 2020 instead of 2010?

---

> ### Author Response · Authors · 2022-11-19
> **Response to Reviewer 4TYn**
>
> Thank you for the valuable review and kind words! We are happy that the contributions on algorithm design (recognizability guided faster sampling) and applications (rectifying bad sketches) are considered novel. We further clarify the raised concerns in the following, and hope it is satisfactory.
>
> > Q1: Similarity to Diff-HW [Luhman and Luhman, 2020]
>
> Similarity generally holds only at a conceptual level – we both train a diffusion model using data in stroke-3 format. We admire [Luhman and Luhman, 2020] for their vision in the early adoption of diffusion models for stroke data.
>
> Ours however differs from them in almost all other aspects (more in the related work):
> - We contribute a novel recognizability-aware sampling strategy for more effective and efficient sampling, whereas Diff-HW simply leverages vanilla DDPM.
> - We introduce a perceptual-metric based gradient guidance to enable conditional sketch generation for sketch refinement and healing.
> - Diff-HW focuses on an entirely different task of text-to-handwriting generation, and uses a different network design (attention block introduced for cross-modal translation). It is therefore not surprising that  [Luhman and Luhman, 2020] performs significantly worse than ours on all metrics (FID, GS, Precision, and Recall as shown in Table 1).
>
> > Q2: The choice of architectures: convolutional U-Net is tailored for dealing with coordinates, and RNN is used to learn the recognizability of sketch.
>
> Good question. Intuitively, using an RNN seems more natural to handle vectorized sketches. However, in our experiments, we empirically found that using a convolutional U-Net for diffusion modeling offers better performance than using an RNN (see the added Table 6 in Appendix A.3). We conjecture this might be a case of a CNN-based framework providing a more holistic positional encoding that better informs the diffusion process (which is also global). RNN is used for recognizability learning simply to avoid the need to synthesize a sketch from the noise output of the U-Net module.
>
> > Q3: Reproducibility
>
> We have now added more implementation details in Appendix A.3. We will release code, build an online demo, and make our pre-trained models publicly available at first instance.
>
> > Q4: Publication year of Diff-HW.
>
> Good spot! This has been fixed in the revision.

---

> > ### Comment · Reviewer_4TYn · 2022-11-20
> > **Thanks for the clarification**
> >
> > Thanks for the clarification and additional experiments!
> >
> > Interesting that CNN-based perform better on coordinates than RNN, I still think it should be explored more throughly but I'm satisfied with the current explanation. And the different from Diff-HW is indeed clear, albeit I still think it's small.
> >
> > Overall, I'm leaning toward acceptance, as the proposed steps skipping is in itself a contribution. I'm keeping my original rating but with more confident toward acceptance.
> >
> > Best,

---

### Official Review · Reviewer_zdyq · 2022-10-25

**Confidence:** 3
**Correctness:** 4
**Technical Novelty And Significance:** 3
**Empirical Novelty And Significance:** 3
**Recommendation:** 8

**Clarity, Quality, Novelty And Reproducibility:**

This work is clearly presented. A good combination of DM and vector data which is not explored before. Looks it is reproducible as authors clearly introduced dataset, hyper-parameters and exp settings in the paper.

**Details Of Ethics Concerns:**

Not aware of ethics concerns

**Strength And Weaknesses:**

+ A nice addition of introducing diffusion model learning into the vector image generation, compare with tons of pixel image generation
+ Joint modeling of both point location and pen state
+ A new adaptive sampling strategy based on sketch recognizability
+ The ablation in Sec. 4.1 is very impressive, covering every aspect about the proposed method so that readers know how each factor will affect the result

The diffusion process in Figure 2 (a) somehow could serve as a tutorial-like process to teach people how to draw an object. I have some minor concerns listed below:

Is this diffusion model designed as class-conditioned or not? Basically, starting from a random scattered points, what type of guidance is driving those points to form the final object, or is it a one2many mapping?

Instead of conditioning on an incomplete sketch to refine, one more interesting application is to invert a given sketch and generate its variations, to a simpler or a better version for different needs.


**Summary Of The Paper:**

This work introduces generating vectorized sketch by modeling the stroke-point locations and pen states via a diffusion model. One major contribution is to embed recognizability of sketch during the sampling. It explores both cases of starting from a random scattered points or an incomplete sketch. Experiments show some appealing results as well as solid quantitative evaluations, ablation study and component analysis.

**Summary Of The Review:**

This is a good extension of applying powerful diffusion models to vector domain which is potentially useful for more complex vector design or art regardless of resolution. I do not have too much criticism towards this work.

---

> ### Author Response · Authors · 2022-11-19
> **Response to Reviewer zdyq**
>
> Thank you for the interesting suggestion on the potential application of our model! We are happy that our attempt to apply the diffusion model to the vector-format image generation is seen as useful.
>
> > Q1: Is the diffusion model designed as class-conditioned or not?
>
> Thanks. Ours is not class-conditional. Given noise input, ours generates a sketch of a random category, i.e., one-to-many mapping. Ours however can be made to be conditioned on a sketch instance, for healing and refinement (sections 3.8 and 4.2).
>
> > Q2: Application of inverting a given sketch into a simpler or a better version for different needs.
>
> Interesting! We have demonstrated that our model can further refine a real human sketch, as validated quantitatively (improved recognition and retrieval accuracy as shown in Table 4, the first row) and qualitatively (Figure 3). However, in our conditional sketch generation, it is not allowed to explicitly control the abstraction level of the generated sketch. We will definitely explore it in our future work.

---

### Official Review · Reviewer_ocgD · 2022-10-27

**Confidence:** 3
**Correctness:** 3
**Technical Novelty And Significance:** 3
**Empirical Novelty And Significance:** 3
**Recommendation:** 8

**Clarity, Quality, Novelty And Reproducibility:**

The paper is well written, with comprehensive visualization and numerical results. The originality of the paper lies in the application of DDPM to the new task, as well as the task-specific fast sampling algorithm.

[Reproducibility] Some of the domain-specific implementation details are lacking, such as how the fixed number of points per sketch are sampled from the dataset, as well as how the guidance scores in the conditional generation tasks are obtained. The paper also did not include any promise of code release.

**Strength And Weaknesses:**

## Strength:
* This is the first paper that applies DDPM to the task of sketch generation.
* The proposed method significantly outperforms previous methods.
* The proposed shortcut sampling scheme significantly reduced sampling time while retaining the quality better than DDIM given the same number of steps.
* The proposed method has all the versatility a DDPM provides -- it can be applied to conditional generation tasks without retraining.

## Weaknesses
* The shortcut sampling technique, although worked well empirically, is not backed by theory. It will be more useful if the author can share more theoretical insights & generalize it to other tasks.
* In section 3.8, the paper mentioned using a pretrained 2D CNN to evaluate similarity between two sketches. It is unclear how the gradient from a 2D CNN is propagated to a vectorized sketch. Is there differentiable rendering?
* The paper used a fixed number of stroke points for each sketch. This might limit its real-world application.

**Summary Of The Paper:**

The paper proposed using diffusion model to generate vectorized sketches. A vectorized sketch is represented as a sequence of points, and a *pen state* indicating whether the pen is touching the paper. The authors formulated the problem as generating a fixed number of 3D points using a 1D convolution-based DDPM. To accelerate the sampling process, the paper proposed using an RNN network to predict whether the noisy example is already recognizable -- empirically, more reverse steps are needed when the result starts to become recognizable.
The proposed model achieved SOTA performance on sketch generation. The version using RNN-based shortcut sampling required significantly less computation compared to Naive DDPM sampling while retaining the generation quality.
In addition, similar to image diffusion models, the proposed model is able to perform tasks such as sketch refinement and healing.

**Summary Of The Review:**

The paper is well-written and the experiments are comprehensive. Being a domain-specific application of an existing technique (DDPM), the paper has not only thoroughly explored the benefits it provides, but also included some interesting new contributions, such as shortcut sampling. Although it will be better if this finding can be generalized beyond sketch generation.

---

> ### Author Response · Authors · 2022-11-19
> **Response to Reviewer ocgD**
>
> Thank you for the insightful review. We are glad that the exploration of applying the diffusion model (with a sketch-specific shortcut sampling strategy proposed) to vectorized sketch generation is considered to be novel and convincing.
>
> > Q1: Theoretical insights behind shortcut samping & if applicable to other tasks.
>
> Thanks. This is a cute discovery we made that is specific to sketch data – coordinates in sketch sequence are far less robust to added noise than pixel values of an image. This can be seen from Figure 1(c) and Figure 5 in Appendix A.4 – that early sampling steps are often long and inefficient. Addressing this merely means a speed-up in generation (16x faster, see Table 1 in the paper), while retaining quality (Table 1).
>
> > Q2: How the gradient from a 2D CNN is propagated to a vectorized sketch
>
> Good catch! There is indeed a differentiable rendering involved, but is common practice in the sketch literature [A,B]. Specifically, we apply neural line rasterization (NLR) [A] to convert a vectorized sketch to a raster image. Essentially, NLR renders line segments between two neighboring stroke points by linearly interpolating their point features. It follows that the obtained 2D image $s_t$ is used to compute the perceptual similarity $p(s_t, s_c)$ to the conditional sketch image $s_c$. Then $p(s_t, s_c)$ will serve as a loss to compute the gradients, which will be backpropagated to each coordinate. Note this rendering process is only needed for conditional generation (i.e., healing and refinement). We have now added a detailed description of the neural line rasterization (NLR) in Appendix A.5.
>
> [A] Li, Lei, et al. "Sketch-R2CNN: An RNN-Rasterization-CNN architecture for vector sketch recognition." IEEE transactions on visualization and computer graphics 27.9 (2020): 3745-3754.
>
> [B] Qi Y, Su G, Wang Q, et al. "Generative Sketch Healing". International Journal of Computer Vision, 2022: 1-16.
>
> > Q3: The fixed number of stroke points might limit its real-world application.
>
> Good point. Admittedly, the number of stroke points N is pre-defined. However, we can see from Table 1 & 3 that except for N=24 which is too short to depict sketches, N=96 works reasonably well to accommodate the different levels of sketch complexity in QuickDraw (the largest human sketch dataset).
>
> > Q4: Implementation details about “how the fixed number of points per sketch are sampled from the dataset, as well as how the guidance scores in the conditional generation tasks are obtained” and reproducibility
>
> We use all the stroke points in order. During training, sketches with more than 96 stroke points are excluded – in reality this is less than 10% of sketches in QuickDraw (we can however set the point number larger as shown in Table 3, if we want to include more training data). For those with less than 96 stroke points, we just zero-pad. Please refer to Q2 for how the guidance scores are obtained for a conditional generation.
>
> We have provided more details on implementation in Appendix A.6. Code will be made publicly available at first instance, together with an online demo.

---

> > ### Comment · Reviewer_ocgD · 2022-12-07
> > **Thanks for the response**
> >
> > Dear authors,
> >
> > Thanks for including the background on the differentiable rendering of line segments.
> >
> > As for stroke point sampling, initially I though some form of resampling is done on the sketches to normalize everything to 96 points -- thanks for the clarification that zero padding is used instead. This does raise the question of whether sketch morphing / interpolation is possible with the proposed model -- the mapping between the underlying sketch representation (fixed number of line segments) and the actual sketch is not a bijection.
> >
> > I maintain my original rating of accept. Applying DDPM to a new domain is invaluable.
> >
> > Best,

---

### Author Response · Authors · 2022-11-19
**General Response**

We thank all reviewers for their time and critical feedback!

We have revised the paper accordingly (marked in blue), and below is a summary:
- more insights on the proposed shortcut sampling are added in Appendix A.4. (Reviewer **ocgD**)
- clarification on how gradients from a pre-trained 2D CNN are backpropagated to a vectorized sketch; a detailed description of this process is now included in Appendix A.5. (Reviewer **ocgD**)
- additional experiments on when the noise predictor is switched from U-Net to bi-directional RNN; this can be found in Appendix A.3. (Reviewer **4TYn**)
- more implementation details have been added in Appendix A.6. (Reviewer **ocgD** and **4TYn**)
- we particularly thank reviewer **zdyq** for proposing a very interesting idea of applying a diffusion model to invert a given sketch to different versions; we have now included discussions on this as potential future work.

---

### Decision · Program_Chairs · 2023-01-20

**Decision:**

Accept: notable-top-25%

**Justification For Why Not Higher Score:**

While this paper is quite nice, it does not present the sort of significantly new machinery or groundbreaking results that would merit bringing to the attention of the whole ICLR community via an Oral. Perhaps if the application domain were something slightly less niche interest, it would be Oral material?

**Justification For Why Not Lower Score:**

The paper achieves quite nice results and shows a potentially non-obvious application of DDPMs. It could inspire others to look at new ways to apply DDPMs to other kinds of sequence data, so it may merit a spotlight presentation.

My "This decision can be bumped down" response to the question following this one means that I am OK with this being downgraded to a Poster-only accept, if the SAC has a difference of opinion. However, I do not think that this paper should be rejected.

Also: paper 261 and this one form a sort of "pair," in that they address similar problems in a similar way (and both achieve good results). If one is accepted, the other should be accepted, else it would be 'scooped.'


**Metareview: Summary, Strengths And Weaknesses:**

This paper proposes a diffusion model to generate vectorized sketches.

Strengths: reviewers noted that this appeared to be the first paper to apply DDPMs to sketch generation. It also produces better results (and produces them faster) than prior methods. Like other DDPMs, it can be applied to conditional generation tasks w/out retraining.

Weaknesses: the method uses fixed number of stroke points for each sketch, which may limit its real-world applicability in some cases.

All reviewers were in favor of acceptance from the start; seems like a clear accept.

**Note From Pc:**

if the above contains the word "oral" or "spotlight" please see: "oral" presentation means -> notable-top-5% and "spotlight" means -> notable-top-25%. As stated in our emails, we are disassociating presentation type from AC recommendations